# Learning from Excellence to Improve Healthcare Services: The Experience of the Maternal and Child Care Pathway

**DOI:** 10.3390/ijerph18041481

**Published:** 2021-02-04

**Authors:** Alice Borghini, Ilaria Corazza, Sabina Nuti

**Affiliations:** Health and Management Laboratory, Institute of Management, Sant’Anna School of Advanced Studies, Piazza Martiri della Libertà, 33, 56127 Pisa, Italy; ilaria.corazza@santannapisa.it (I.C.); sabina.nuti@santannapisa.it (S.N.)

**Keywords:** best practices, performance management systems, maternity pathway

## Abstract

The ability to deal with adversity and the resilience of people and groups are shown to depend positively on the tendency to nurture positivity. Therefore, the aim of this study is to evaluate whether Learning from Excellence (LfE) can be an effective method to manage systematic health systems, when transparent disclosure and benchmarking of data are adopted in performance evaluation. This study consists of a quantitative and a qualitative phase. In the former, maternal care is investigated at the regional level, starting from performance data and indicators of the maternity pathway referred to 98 healthcare providers in 10 Italian regions, that share the same evaluation system. The second phase investigates qualitatively the organizational determinants and the experience of professionals involved in the pathway, through the organization of on-site workshops. We identified the seven best practices among the 42 units of analysis. Communication, trust and shared goals among health professionals involved in the pathway emerged as core themes from the qualitative analysis. This study confirms that LfE under the conditions of benchmarking assessment and transparent disclosure of data can be implemented systematically in management practice, in order to boost health personnel’s resilience and, in general, the organizational climate in the working environment.

## 1. Introduction

According to a number of studies conducted in psychology and economics, human beings have an innate tendency to (1) be affected or notice negative occurrence [1,2] and (2) value loss more than gain [3]. It was demonstrated that such predisposition to “negativity bias” is also common to healthcare professionals, mainly because of canonical training and education in medicine [4]. However, recent studies have showed that individuals learn by reflecting either on failure (negative reinforcement) or on success (positive reinforcement) [5]. Moreover, involving health professionals was demonstrated to be directly associated with cultivating positivity in individuals and teams, and improved resilience and the ability to deal with adversity [6].

Indeed, a commonly adopted approach in the healthcare sector is the positive deviance (PD) approach, intended to focus on unstandardized performance and process variations that lead to positive results, making leverage on the diffusion of positive examples within healthcare organizations [1]. Such a process, in fact, constitutes the real engine of the Learning from Excellence (LfE) model, as proposed by Kelly and colleagues (2016). Indeed, the LfE model can be defined as a management tool in that it seeks to foster learning among peers starting from episodes of excellence, based on the idea that everyday successes are as valuable as defects. The possibility of increasing learning while cultivating positivity [2] as well as excellence proves the ability of healthcare institutions to be strongly aligned with the ultimate goals of the health system.

Additionally, the public and transparent dissemination of results of the performance evaluation systems is particularly relevant in the health sector, for at least two reasons. First, since resources in the public sector are scarce and mainly provided through general taxation, there is a substantial need for the accountability of policy-makers and managers. Secondly, based on the concept of professional bureaucracy given by Mintzberg in 1978, the nature of healthcare is often centered on professional figures with high skills and wide margins of autonomy [3]. These characteristics make the reputational lever an essential tool for enhancing health professionals’ work [4,5,6].

Since the reputational lever and the positive reinforcement approach are important in healthcare, it could be recommended to focus on what goes well without excluding the failures in the healthcare system. Therefore, the LfE model can play an important function in the performance management of the healthcare sector, hence the opportunity to spread such a model as a governance tool across various levels in the system [7].

The present study investigates how the LfE model can be introduced in healthcare in a systematic and integrated way as an innovative tool to manage performance and boost the motivation of health professionals in order to improve health services [2,8,9]. More particularly, this paper aims at answering the following research question: can the LfE model be a management tool to improve healthcare system performance, when based on a benchmarking assessment process and reinforced by a reputation leverage?

In practical terms, the authors describe an attempt to apply the LfE model in the maternity pathway, identifying the best practices among 98 healthcare providers located across 10 Italian regions that share an interregional performance evaluation system (Italian Regional Performance Evaluation System—IRPES) [10,11].

## 2. Materials and Methods

This paper illustrates the results of a mixed-methods study carried out in nine Italian regions and two autonomous provinces sharing a voluntary-based Italian Regional Performance Evaluation System (IRPES) [12,13,14].

The IRPES consists of both observation and evaluation indicators, evaluated based on quintiles. The evaluation reference standards come from the national and international literature and, where lacking, from the median of the various providers in the IRPES [15].

Following the standards identified, evaluation scores ranging from 0 to 5 are then calculated for each indicator, where 0 corresponds to a poor performance while 5 corresponds to a very good performance. The IRPES tries to overcome the logic of evaluation by healthcare settings, in order to value every step along the patient pathway across different providers and services [14].

The researchers focused on the maternity pathway that counts 15 indicators overall, subdivided into 3 phases: delivery, first year of the newborn and pediatric phase. In order to spread a population medicine approach, the researchers selected a set of different types of indicators (i.e., process and outcome indicators) that represent different levels of analysis (e.g., hospital- and local health authority (LHA)-level). The selection of the indicators was carried out according to the most recent scientific literature [16,17].

Table 1 shows all indicators of each phase of the pathway.

To diffuse the LfE model along the maternity pathway:Quantitative information from IRPES was used to assess maternal care among different regional healthcare systems.A qualitative approach was adopted to investigate the organizational models and tools used, as well as the perspective of key professionals involved in the pathway.Experiences, management models and data were shared across a professional community to spread the best practices model throughout the IRPES.

The project started in 2018 and was conducted over a period of about eight months.

Firstly, in the quantitative phase, the researchers used the 15 indicators of the pathway to analyze nine Italian regions and two autonomous provinces using a modified positive deviance model, which identified the best performers among geographical areas (GAs) including both local health authorities (LHAs) and teaching hospitals (THs) which reside in the same territory.

The weighted average of the evaluations of the single indicators for each phase of the pathway gives the phase evaluation at the provider level. In turn, the weighted average of the phase evaluations of the providers gives the phase evaluation at the GA level.

Moreover, the arithmetic mean of the phase evaluations of the pathway at the GA level gives the overall evaluation of the pathway at the GA level.

The final result of each GA is a continuous value in the 0–5 range [14].

After computing all pathway evaluations at the GA level, the researchers ranked these evaluations and identified the best performers as all the GAs whose evaluations were higher than 3 in the evaluations distribution.

The qualitative section of this article was written adapting the COnsolidated criteria for REporting Qualitative research (COREQ) for reporting qualitative research [18].

The research group consists of research fellows and Ph.D. students in healthcare management, having different backgrounds (economics, medicine, sociology and political sciences).

The participants in the workshops were employed by healthcare institutions sharing the IRPES and, even if not directly in touch with researchers, all of them were aware of the ultimate purposes and rationale of this study.

The researchers selected one best performer for each regional healthcare system from the group of best performing Gas, and they conducted one workshop in collaboration with the selected best performer to detect the clinical and organizational model beyond the successful performance. In the Veneto region, according to the regional governance, the second-best performer was selected because of the exceptional logistical characteristics of the first-best performer, Venezia area, which is rarely comparable with the other Italian and international regions.

The decision to organize on-site workshops to collect information and celebrate best practices was motivated by the fact that storytelling workshops are demonstrated to be an effective method to facilitate organizational learning [19].

Moreover, such a decision in methodological terms is based on the most relevant findings of social norms and self-determination theory, according to which, the use of non-monetary incentives, as well as image and reputation, play a fundamental role in promoting pro-social behavior [20,21].

When organizing the workshops, to ensure the comparability, the healthcare providers were asked to involve all stakeholders who play a relevant role in the maternity pathway (i.e., gynecologists, pediatricians, obstetricians, nurses, administration staff and clinical directors) and to communicate to participants specific guidelines to be followed for their presentations: taking charge of the pregnant woman during pregnancy, management of childbirth assistance in hospitals, taking care of the mother after childbirth, taking care of the newborn in the first year of life, taking care of the pediatric child, and transversal aspects [22].

The workshops took place in the headquarters and lasted around three hours. The average number of participants in the workshops were 17 (minimum 8–maximum 28) plus two or three researchers. They were organized to facilitate health professionals in providing rich and extensive information on the organizational model and tools adopted, as well as detailed information on their working environment. In addition, visits were planned to create a trusting relationship between the researchers and clinicians, in order to openly discuss their practices.

In order to perform analyses, the researchers recorded all presentations given by health professionals, wrote them down and used the notes taken during the workshops. Starting from the collected materials, an anecdotal analysis was conducted separately by two of the researchers involved, following a predefined check-list.

First, there is a description of the organization regarding the features of external and internal contexts where the provider is embedded.

Particularly, the external context includes aspects related to the socio-demographic and clinical characteristics of the LHAs reference population, and geographic peculiarities of the LHAs reference GA.

The internal context, instead, considers the number of hospitals managed by the LHAs, either public or private, the number of THs where available, the number of employees and beds, and the structure of primary care as well as general performance data.

Second, the analysis of clinical and organizational determinants, intended to identify the specificities of the pathway structure and activities (i.e., analysis of the performed activities, involved professionals and functions carried out in each phase of the pathway), the operational mechanisms within the organization (i.e., computation and evaluation of indicators, and priority-setting) and the management of the pathway (i.e., interactions among professionals, communication strategies and trust).

Third, the identification of strengths beyond the successful performance of the provider based on the explanations of health professionals during the workshops, according to the scientific literature and clinical guidelines, and their transferability onto other realities of the system.

Finally, the project aimed at creating a community of professional practice to share experiences, management models and data in order to spread the best practice models across the IRPES. The process of engaging clinicians was based on the methodology of action research [23,24,25].

In particular, it involved in the workshops not only the clinicians who play a key role in the maternity pathway, but also apical figures coming from other geographical areas.

Researchers then collected all of the information from the workshops in a short book, which was launched during a plenary meeting with managers, representatives and clinicians of the IRPES Network, in order to spread the best practice’s approach among other participants as well as to discuss and compare different realities.

## 3. Results

Among the 10 investigated Italian Regions, there are 42 units of analysis embedded within nine regional healthcare systems and two health systems belonging to autonomous provinces, in Trento and Bolzano, respectively.

The 42 units of analysis correspond to different GAs within participating regions of the IRPES, where each GA considered may include both LHAs and THs. The ranking of GAs identified overall seven best performers that are those whose evaluation scores lie above the value of 3 in the scores distribution (≥3.01). According to the ranking, the first best performer result was 3.43, while the worst performer reached value 1.46. Table 2 illustrates the ranking of the seven best performers (area evaluation) with respective overall evaluations by provider (provider evaluation). The evaluation scores of each phase of the pathway are given both at the GA and provider level. In some phases of the pathway, a lower number of evaluation scores is available for single providers because of the different delivery setting.

Among the identified best performers, only four belonging to different regions were invited to join the on-site workshops (highlighted in Table 2). The results of the quantitative data analysis were shared and discussed with the regional managers to decide on which GA belonging to each best performing region to focus, in order to have a perspective on the peculiarities of each involved regional healthcare system.

The results arising from the qualitative analysis of these GAs showed that communication, trust and shared goals among health professionals played a key role among the best performing hospitals and LHAs.

All best performers adopt the pathway as an integration, coordination and sharing tool of clinical and organizational models.

The existence of shared organizational procedures, through the diffusion of technical protocols within the organization, determines a clear definition of functions to be dispatched by various healthcare professionals along the pathway, and among different providers.

Another key issue was a clear-cut role definition of professionals along the pathway, as well as the multidisciplinary and multi-professional (or profile) working environment.

In terms of management of the pathway, the interviewed best performers organize regular internal meetings among multi-profile health professionals and implement specifically tailored communication strategies with respect to the different phases of the pathway (i.e., by means of IT support, regular internal conventions and meetings, etc.).

Regarding the operational mechanisms, the best performers have an internal performance measurement and evaluation system, based on the computation of both clinical and organizational indicators, not necessarily imposed by the general director of the GA.

Thanks to the evaluation of these results, the hospital management can establish step-by-step how to achieve priorities and objectives set according to the hospital mission and strategic plan.

A crucial role in the management of the pathway is played by the detection of existing mutual trust across service users and providers, as well as different categories of health professionals, throughout the diverse phases of the pathway.

For further detail, Table 3 shows a selection of relevant and typical quotes of some health professionals involved in the on-site workshops, with the respective dimensions. The most important traits emerging from the anecdotal analysis of all quotes collected are the following:There are periodical meetings of health professionals having a different profile and specialization, which facilitates the integration of different health services along the pathway (integration).It emerges a very centralized management in terms of both organizational and clinical guidelines (centralized management).In all realities, a clear-cut definition of roles and responsibilities is provided for each health professional involved in the pathway (role definition).There is a significant attitude towards direct and massive involvement and information of mothers and families across the various pathway phases (involvement and information).Special attention is drawn to the delivery of personalized care according to mothers’ specific needs and health conditions (personalized medicine).Processes are addressed according to major and up-to-date findings in the national and international literature (evidence-based medicine).

## 4. Discussion

The adopted methodology supports the introduction of the LfE model in healthcare as a tool to improve the performance of maternity services by motivating health professionals [2,8,9,26]. The innovation of our work consists of three aspects: the identification of the best practices based on a quantitative data analysis process, the qualitative analysis of the organizational determinants beyond excellent performances and the celebration of health professionals directly involved in the maternity pathway [19,20,21].

Regarding the results of the quantitative analysis, in methodological terms this study contributes to the development of previous examples on the application of the positive deviance approach [1,27,28], in particular because the researchers adapted the definition of excellence to a collective, pluralist, integrated and iterative process. Moreover, the concept of best practice was also revised, together with the identification process. First, the methodology presented here covered the full range of available performances, considering a variety of indicators relative to the maternity pathway as a whole [17] even though, in order to provide a full view of the maternity pathway, indicators pertaining to the pregnancy phase should also be included in further studies. Secondly, the selection and definition of geographic reference areas was undertaken to accurately identify the best performers in taking care of mothers and children, therefore including health and social services offered at either hospital or residential level.

Concerning the results of the qualitative analysis, the most significant contribution of this study relates to the evidence emerging from the investigation of the organizational determinants beyond the successful performance [7]. Through the anecdotal analysis, several key aspects arose as being shared by the best performers. More particularly, the most relevant findings are that (1) a strong attention is drawn onto services users, both mothers and children, (2) a significant investment is faced on health professionals and, in general, on human resources and (3) a well-defined managerial orientation towards the development of a robust organizational architecture is aimed at supporting integration processes among health professionals. Additionally, it is worth mentioning that these key aspects result in being embedded in a systemic view and they are mutually affected by positive interpersonal interactions. Indeed, they all work in synergy and somehow constitute the pillars of a permanent virtuous cycle.

Another crucial aspect underlying the qualitative analysis regards the process of celebration and merit enhancement of health professionals pursued by means of the onsite workshops [19,20,21]. The health professionals involved in the maternity pathway were invited to illustrate their own work and activities to the researchers, and top managers from other Italian healthcare institutions were invited to join the workshops. Moreover, these workshops were devoted to the celebration of health professionals, since the reputational capital is known to be one of the most effective leverages of motivation of health professionals in the healthcare sector [4,25].

Therefore, the results suggest that all of the best performers are characterized by efficient and effective care organization and not only health professionals’ clinical competencies, hence scope for changes may become more straightforward and actionable.

Finally yet importantly, an attempt was made to identify the determinants of success, so as to model them in order to become replicable. In the health care sector, it is common practice to organize internal audits to investigate and analyze negative phenomena by creating checklists and reference protocols. However, as explained by Hollnagel et al. in 2015, the proportion of negative events is much smaller than positive occurrences even in complex situations. From this, the opportunity to implement the same process for positive phenomena should be evaluated, in order to model and regularly spread excellent performances throughout healthcare organizations [29].

## 5. Conclusions

The identification and celebration of the best performers through the implementation of the LfE model holds the potential to promote the improvement of processes and activities among healthcare providers. This potential is also unlocked in the maternity pathway management field, making the sharing of the best performers’ experiences possible at both a regional and national level. The methodology, as it was adapted in the present study, plays the same role of the policies and practices enabling interactions among healthcare workforce and patients, ultimately representing the “integration of care” [7].

A further validation of the proposed model implies enlarging time and geographic constraints as well as dealing with clinical and organizational limitations. Considering the results in trend is necessary in order to confirm these findings and eventually identify other realities capable of great improvements, even if lacking an excellent performance. Moreover, it would be advisable to extend this kind of analysis also to other pathways.

The framework used here to anecdotally analyze the information collected during on-site workshops could be adapted in order to conduct a content analysis based on pre-fixed parameters.

Finally, the study conducted and illustrated so far in the present article may have mainly a narrative value, therefore it would be interesting to evaluate the impact of our intervention in developing a specific evaluation tool, such as a survey, to be administered to the best performers.

In conclusion, it should be further investigated as to how health professionals can learn systematically from positive results, in order to model the behavior adopted to achieve the excellent results. In other words, modeling the best performers’ behavior is the key element to guarantee the standardization of the process in order to replicate it in another context.

## Figures and Tables

**Table 1 ijerph-18-01481-t001:** Maternity pathway indicators for evaluation.

Pathway phases:	Indicators:
Delivery	Percentage of NTSV * caesarean sections
	Percentage of NTSV episiotomies
	Percentage of operational births (using forceps or sucker)
First year of newborn’s life	Hospitalization rate per 100 inh. (≤1 year)
	Vaccination coverage (measles, mumps and rubella)
	Vaccination coverage (meningococcal)
	Vaccination coverage (pneumococcal)
	Vaccination coverage (hexavalent at 24 months)
Pediatric age (>1)	Vaccination coverage (varicella)
	Hospitalization rate per 100 inh. (>1 year)
	Hospitalization rate for asthma per 100.000 inh. (2–17 years)
	Hospitalization rate for gastroenteritis per 100.000 inh. (0–17 years)
	Hospitalization rate for tonsillectomy per 100.000 inh.
	Antibiotics consumption
	Cephalosporins consumption

* Nulliparous, Term, Singleton, Vertex caesarean sections.

**Table 2 ijerph-18-01481-t002:** Ranking of best performers by GA and provider extracted from the results of the quantitative phase.

Ranking	Region	Area	Provider	Area Evaluation	Provider Evaluation	Delivery Phase (Area)	Delivery Phase (Provider)	First Year of Newborn’s Life (Area)	First Year of Newborn’s Life (Provider)	Paediatric Age (>1) (Area)	Paediatric Age (>1) (Provider)
1	Lombardy	Bergamo	ASST * Papa Giovanni XXIII	3.43	4.21	4.05	4.21	2.92		3.31	
1	Lombardy	Bergamo	ASST di Bergamo Est	3.43	3.77	4.05	3.77	2.92		3.31	
1	Lombardy	Bergamo	ASST di Bergamo Ovest	3.43	4.21	4.05	4.21	2.92		3.31	
1	Lombardy	Bergamo	ATS ** Bergamo	3.43	3.12	4.05		2.92	2.92	3.31	3.31
2	Veneto	Venezia	Azienda ULSS n. 3 Serenissima	3.39	3.39	3.48	3.48	2.67	2.67	4.01	4.01
3	Veneto	Marca Trevigiana	Azienda ULSS n. 2 Marca Trevigiana	3.25	3.25	3.68	3.68	2.08	2.08	3.99	3.99
4	Lombardy	Montagna	ASST della Valcamonica	3.25	4.57	3.99	4.57	3.15		2.61	
4	Lombardy	Montagna	ASST Valtellina Alto Lario	3.25	3.47	3.99	3.47	3.15		2.61	
4	Lombardy	Montagna	ATS della Montagna	3.25	2.88	3.99		3.15	3.15	2.61	2.61
5	Lombardy	Brianza	ASST di Lecco	3.23	2.88	3.26	2.88	3.25		3.18	
5	Lombardy	Brianza	ASST di Vimercate	3.23	3.61	3.26	3.61	3.25		3.18	
5	Lombardy	Brianza	ASST di Monza	3.23	3.02	3.26	3.02	3.25		3.18	
5	Lombardy	Brianza	ATS della Brianza	3.23	3.21	3.26		3.25	3.25	3.18	3.18
6	AP Trento	Trento	APSS *** Trento	3.23	3.23	3.94	3.94	2.20	2.20	3.54	3.54
7	Tuscany	Centro	AUSL **** Centro	3.01	3.08	3.19	3.41	2.41	2.41	3.42	3.42
7	Tuscany	Centro	AOU ***** Careggi	3.01	2.68	3.19	2.68	2.41		3.42	

*Azienda Socio-Sanitaria Territoriale; **Agenzia di Tutela della Salute; ***Azienda Provinciale per i servizi sanitari; ****Azienda Unità Sanitaria Locale; ***** Azienda Ospedaliera Universitaria.

**Table 3 ijerph-18-01481-t003:** Selection of representative quotes and identified dimensions from the anecdotal analysis.

GA (Region)	Quotes	Dimensions
**Bergamo (Lombardy)**	“It is important to bring around a table all those interested in protecting health”	Integration
	“We have a single clinical direction, which allows homogeneity and affects the performance of all”	Centralized management
	“A lot of informative material is provided to mothers and families; we produced booklets and brochures”	Involvement and information
**Marca Trevigiana (Veneto)**	“By reviewing the literature, we found out that two caesarean sections increase the risk of uterine rupture by 0.1%, so we encourage women to have a vaginal delivery even if they had a previous cesarean section”	Evidence-based medicine
	“We meet monthly with future parents who enter the hospital, where we present the ward and professionals involved in the pathway—neonatologist, anesthesiologist, obstetrician and gynecologist—showing them the services in the facility”	Involvement and information Role definition
	“The ambition is to try to standardize procedures and consents as much as possible. It is not easy to meet as we are six hospitals and the functional department also encompasses the territory. But I think the future is to share resources”	Centralized management
**Trento (A.P. Trento)**	“Taking care of mothers is precisely defined on the responsibilities of the midwife”	Role definition
	“The focus of the pathway is not only on the pregnant woman, but on the concept of parenting”	Involvement and information
	“We focus on the concept of differentiated management of patients, for intensity of control and care, to be able to send the patient in the most appropriate setting for her clinical situation”	Evidence-based medicine Personalized medicine
**Centro (Tuscany)**	“It is important to identify paths based on the degree of risk”	Personalized medicine
	“The goal of our organization has always been to put the couple with the child at the center”	Involvement and information
	“An added value is given by the multidisciplinary sharing of this pathway”	Integration Role definition

## Data Availability

Publicly available datasets were analyzed in this study. This data can be found here: [http://performance.sssup.it/netval/start.php].

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
