# Peer review of "Learning from Excellence to Improve Healthcare Services: The Experience of the Maternal and Child Care Pathway"

_ijerph, 2021, doi:10.3390/ijerph18041481_

Round 1
Reviewer 1 Report
Thank you for the opportunity to review your manuscript. This study tried to examine whether the Learning from Excellence model based on a benchmarking assessment process can become a management tool of the healthcare system. Providing a management tool for the healthcare system is a significant issue, however, the manuscript has some concerns.
- The authors illustrated the things concerning the LfE model in the introduction section. However, it was complicated. Therefore, it was difficult to understand the LfE model. In particular, the definition of excellence was unclear. Please, the definition of the LfE needs to describe.
- In the materials and methods section, you provide the 15 items as the indicators. Please, the validity of these items should be provided.
- In the materials and methods section, you need to provide how to select one best performer for each Regional Healthcare System (L113). you should make clear the definition of the group of best performing Geographical Areas.
Author Response
Point 1: The authors illustrated the things concerning the LfE model in the introduction section. However, it was complicated. Therefore, it was difficult to understand the LfE model. In particular, the definition of excellence was unclear. Please, the definition of the LfE needs to describe.
Response 1: We thank the reviewer for the observation. In the introduction of the manuscript, we tried to explain better what is intended by Learning from Excellence (LfE) model – for more detail see lines 41-49.
Point 2: In the materials and methods section, you provide the 15 items as the indicators. Please, the validity of these items should be provided.
Response 2: We thank the reviewer for the comment. The selection of the indicators was conducted according to the most recent scientific literature (Nuti S, Bonini A, Murante AM, et al. Performance assessment in the maternity pathway in Tuscany region. Heal Serv Manag Res 2009;22:115–21. doi:10.1258/hsmr.2008.008017; Escuriet R, White J, Beeckman K, et al. Assessing the performance of maternity care in Europe: A critical exploration of tools and indicators. BMC Health Serv Res 2015;15:1–13. doi:10.1186/s12913-015-1151-2).
For more detail, we invite you to view the calculation sheets of the selected indicators: https://performance.santannapisa.it/pes/start/start.php.
Point 3: In the materials and methods section, you need to provide how to select one best performer for each Regional Healthcare System (L113). you should make clear the definition of the group of best performing Geographical Areas.
Response 3: We thank the reviewer for the note. We tried to address this comment in the Materials and Methods section of the manuscript through lines 131-137.

Reviewer 2 Report
The topic is interesting but not suitable for international readers. The study is only descriptive, without any comparative analysis or establishing possible associations. This great weakness is not addressed in the discussion.
Need more background information to demonstrate the knowledge gap and rationales of the proposed study.
The objective is not clearly stated
The current wording of the article makes it difficult for the reader to understand.
Most of the bibliographic references are older than 10 years, there are even some older than 40 years.
Author Response
Point 1: The topic is interesting but not suitable for international readers. The study is only descriptive, without any comparative analysis or establishing possible associations. This great weakness is not addressed in the discussion.
Response 1: We thank the reviewer for the observation. We tried to address this comment as a limitation and opportunity for future research in the conclusions of the manuscript (for more detail see lines 441-447).
Point 2: Need more background information to demonstrate the knowledge gap and rationales of the proposed study.
Response 2: Some more bibliographic references were included in the introduction of the manuscript (Bevan G, Fasolo B. Models of governance of public services: Empirical and behavioural analysis of ‘econs’ and ‘humans’. In: Behavioural Public Policy. Cambridge: : Cambridge University Press 2013. 38–62. doi:10.1017/CBO9781107337190.003; Codling S. Best Practice Benchmarking: A Management Guide. 2nd ed. Gower Pub Co 1995; Bullivant JR. Bullivant, J. R. (1998). 1st ed. Urch Publishing Ltd 1998.).
Point 3: The objective is not clearly stated.
Response 3: We tried to make the objective of our research clearer, by rephrasing the research question in the introduction of the manuscript (for more detail see lines 64-71).
Point 4: The current wording of the article makes it difficult for the reader to understand.
Response 4: Thank you for your notice. We are going to make a request for a proof read by a mother tongue speaker.
Point 5: Most of the bibliographic references are older than 10 years, there are even some older than 40 years.
Response 5: Thank you for checking carefully the bibliography of our research. We confirm that some of the references included are quite old but we would like to keep them because they are classics of the international literature in management and performance evaluation. However, if there is any particular reference you have in mind, we will be very happy to include your suggestions and advises to improve our work.
